# Genetic Characteristics and Microbiological Profile of Hypermucoviscous Multidrug-Resistant *Klebsiella variicola* Coproducing IMP-4 and NDM-1 Carbapenemases

Bingjie Wang,[a] Fen Pan,[a] Dingding Han,[a] Wantong Zhao,[a] Yingying Shi,[a] Yan Sun,[a] Chun Wang,[a] Tiandong Zhang,[a] (ID) Hong Zhang[a]

[a]Department of Clinical Laboratory, Shanghai Children's Hospital, Shanghai Jiaotong University, Shanghai, China

Bingjie Wang and Fen Pan contributed equally to this article. Author order was determined both alphabetically and in order of increasing seniority.

**ABSTRACT** We report here a hypermucoviscous, New Delhi metallo-$\beta$-lactamase 1 (NDM-1) and imipenemase 4 (IMP-4) carbapenemases-coproducing *Klebsiella variicola* isolate obtained from a pediatric patient. This strain was resistant to carbapenems and most other $\beta$-lactams. Although hypermucoviscous, this strain possessed attenuated virulence according to serum killing assay and *Galleria mellonella* infection model. Notably, two copies of $bla_{NDM-1}$ were contained on two tandem IS*CR1* elements and coexisted with $bla_{IMP-4}$ in a novel hybrid multidrug resistance plasmid. This is the first description of the coexistence of $bla_{NDM-1}$ and $bla_{IMP-4}$ in a single plasmid of hypermucoviscous *K. variicola*.

**IMPORTANCE** As an important member of the *Klebsiella pneumoniae* complex, *Klebsiella variicola* is poorly studied as an emerging human pathogen. We, for the first time, report a unique *K. variicola* isolated from a pediatric patient in China. This isolate exhibited hypermucoviscosity, a classic hypervirulence characteristic of *K. pneumoniae*, and contained multiple carbapenem-resistant genes, including $bla_{IMP-1}$ and $bla_{NDM-1}$. Interestingly, these antimicrobial resistance genes were located on a novel hybrid plasmid, and our results suggested that this plasmid might have been introduced from *K. pneumoniae* and undergone a series of integration and recombination evolutionary events. Overall, our study provides more insight into *K. variicola* and highlights its superior capability to acquire and maintain foreign resistance genes.

**KEYWORDS** IMP-4, IS*CR1*, *Klebsiella variicola*, NDM-1, carbapenemase, hypermucoviscous, whole-genome sequencing

Carbapenem resistance in *Klebsiella pneumoniae* represents a serious threat to human health worldwide, and it is driven mostly by the dissemination of carbapenemase, largely composed of *Klebsiella pneumoniae* carbapenemase (KPC), New Delhi metallo-$\beta$-lactamase (NDM), and imipenemase (IMP) (1). *Klebsiella variicola*, a subset of the *K. pneumoniae* complex, is a commensal bacterium capable of colonizing plants, animals, and humans (2–4). However, it has been considered an emerging pathogen in humans due to recently gaining recognition as a cause of blood and urinary infections (4). *K. variicola* isolates generally display lower antibiotic resistance rates than *K. pneumoniae* (4). Although being increasingly reported worldwide, *K. variicola* as an infectious agent in humans is poorly studied (5–8).

In 2018, *K. variicola* strain SHET-01 was recovered from the urethral catheter of a 2-year-old female patient in the pediatric intensive care unit of a teaching hospital in Shanghai, China. Based on the string test, this strain displayed hypermucoviscosity (HMV), indicating that it is highly likely to be hypervirulent (9). Antimicrobial susceptibility testing showed that this strain exhibited resistance to the carbapenem antibiotics meropenem, imipenem, and ertapenem, with MICs of 32, 32, and 16 $\mu$g/mL, respectively

**Ad Hoc Peer Reviewer** (ID) Le Van Chuong, University of Medicine and Pharmacy at Ho Chi Minh City

Address correspondence to Hong Zhang, schjyk2015@126.com.

The authors declare no conflict of interest.

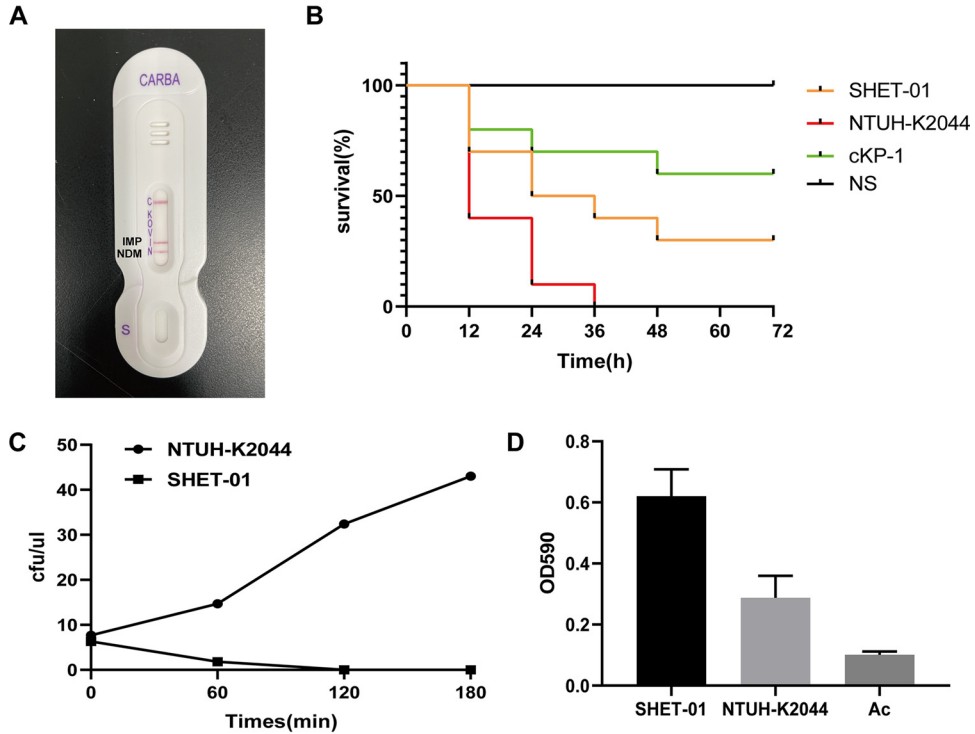

**FIG 1** The microbiological phenotype of *K. variicola* strain SHET-01. (A) The production of NDM and IMP carbapenemases in SHET-01 detected on an NG-Test CARBA 5. (B) Virulence potential of *K. variicola* SHET-01 in the *G. mellonella* infection model. NTUH-K2044 was used as a high-toxicity control, cKP-01 as a low-toxicity control, and NS as a blank control. Mantel-cox test was used for statistical analysis of the survival curve; *, $P < 0.05$. (C) Serum killing assay. (D) Analysis of biofilm formation; Ac, the negative control. The biofilm formation scores of SHET-01 and NTUH-K2044 were 6.13Ac and 2.84Ac, respectively.

(Table S1). Consistent with this, the coproduction of NDM and IMP carbapenemases was confirmed by NG-Test CARBA 5 lateral flow assay (Fig. 1A). Considering the clinical significance of hypervirulence and the coproduction of dual carbapenemases, here we report the genetic characteristics and microbiological traits of this unique strain, providing new insight into *K. variicola*.

We performed whole-genome sequencing using PacBio Sequel single-molecule real-time (SMRT) sequencing platform and the Illumina NovaSeq sequencing platform (paired-end, 2 by 150 bp) to investigate the genomic feature of this isolate. The isolate harbored a circular chromosome of 5,508,387 bp and two plasmids, pNDM-IMP-1 (347,317 bp) and pNDM-IMP-2 (144,371 bp). This strain was assigned to ST3936 based on multilocus sequence typing (MLST) (http://bigsdb.pasteur.fr/klebsiella/klebsiella.html) and KL30 by Kaptive (https://github.com/katholt/Kleborate/). Comparative analysis of the chromosome of SHET-01 with *K. variicola* DX120E revealed a highly similar genome content (99.9% identity and 96% query coverage), confirming that SHET-01 belonged to *K. variicola* (Fig. S1).

As shown in Table S2 and Fig. S1, the chromosome of the *K. variicola* SHET-01 possessed a number of canonical *K. pneumoniae* virulence factors, including enterobactin (*entABCDEFS*), salmochelin (*iroN*), aerobactin (*iutA*), the KFU iron uptake system (*kfuABC*), fimbria type 1 (*fimABCDEFGHI*), and fimbria type 3 (*mrkABCDFIJ*) (10). Additionally, biofilm, urea, and other pathogenicity factors also exist in this strain. However, the *rmpA/rmpA2* or *magA* genes typical of the HMV phenotype of *K. pneumoniae* were not found, implying the existence of *rmpA/rmpA2*-independent HMV pathways in *K. variicola* (11). *In vitro* and *in vivo* virulence assays showed that strain SHET-01 was capable of forming a strong biofilm but exhibited a high degree of susceptibility to human serum and a virulence level significantly lower than that of the hypervirulent control strain NTUH-K2044, indicating that strain SHET-01 is phenotypically hypovirulent (Fig. 1B to D).

Additionally, this strain harbored numerous classes of antimicrobial resistance genes, such as aminoglycosides, fluoroquinolones, carbapenems, and other $\beta$-lactams (Table S2). The carbapenemase-encoding genes, $bla_{NDM-1}$ (two copies) and $bla_{IMP-4}$, were colocated on plasmid pNDM-IMP-1. This plasmid was a 347.3-kb large untypeable plasmid, containing most of the resistance genes, except for the intrinsic *oqxAB* and $bla_{LEN17}$ (Fig. 2A, Fig. S1). A BLASTN search showed that pNDM-IMP-1 exhibited 100% identity and 83% query coverage with pKP1814-1 (KX839207.1), a $bla_{IMP-4}$-harboring plasmid of *K. pneumoniae* strain isolated from an outpatient in Hebei province (12). Comparative analysis revealed that pNDM-IMP-1 and pKP1814-1 shared similar plasmid backbones that contained highly conserved plasmid functional genes (such as replicon gene *repA* and stability genes *telB-stabD*), conjugative transfer gene clusters, insertion elements, drug resistance determinants (such as $bla_{IMP-4}$ and $bla_{SFO}$), and other functional genes but had an ~70 kbp unique genetic makeup. A further BLAST search of this unique genetic region showed that it shared homology with p2315-2-NDM (CP039829.1) and BHW35 unnamed plasmid (CP020508.1) (25% coverage, 99.76% identity), with several fragments remaining unidentified (Fig. 2A), suggesting that genetic recombination events and distinct DNA mobility events might be responsible for the generation of pNDM-IMP-1.

Similar to the previously reported pKP1814-1 plasmid, $bla_{IMP-4}$ was also carried by a novel composite transposon Tn6404, which was flanked by IS5075, and contained an In804-like integron (Int1-$bla_{IMP-4}$-*itrA-qacG2-aadA4-catB3*) located downstream of the *mer* operon (*merRTPFADE*) together with the Tn5563a-like transposon (*tinR-tnpA-merT-merR*) and upstream of the remnant of transposon Tn1696 (*tnpR-tnpA*) (Fig. 2B). Studies have revealed that $bla_{IMP-4}$-containing In809 integron (Int1-$bla_{IMP-4}$-*qacG2-aacA4-catB3*) was globally circulating among *Acinetobacter* and *Enterobacteriaceae* (13). The In809-like integron was identical to the In809 integron, except for insertion of *itrA*. Notably, pNDM-IMP-1 and pKP1814-1 shared this region as mentioned above, which indicated the transmission potential of $bla_{IMP-4}$-containing In809-like integron in *Klebsiella* species.

Furthermore, we found that two copies of $bla_{NDM-1}$ were carried by two tandem copies of insertion sequence common regions 1 (ISCR1) element (ISCR1-ISAba125$\Delta$-$bla_{NDM-1}$-$bla_{MBL}$-*trpF-dfrA27-aadA16-qacE$\Delta$1-sul1*), which was similar to the two tandem copies of $bla_{NDM-1}$-ISCR1 element of the chromosome of *Pseudomonas aeruginosa*, except that pNDM-IMP-1 existed in the absence of *aphA6*, the truncation of ISAba125, and the insertion of *trpF-dfrA27-aadA16* (14). Downstream of ISCR1 element is an Int1 integron and six resistance cassettes (*aadA4-$bla_{OXA-1}$-catB3-aar-3-qacE$\Delta$1-sul1*), which was similar to the int1-ISCR1 structure of pEcNDM1 from *Escherichia coli*, except that pNDM-IMP-1 existed in the absence of *dfrA27* (Fig. 2C) (15). The ISCR1 element is a type of insertion sequence and a powerful gene-capturing tool that can mobilize antibiotic resistance genes and provide a promoter for the expression of nearby genes (16). Our data suggested that the ISCR1 element might play an important role in the mobilization of $bla_{NDM-1}$ and the recombination of multiple resistance region.

To the best of our knowledge, this is the first report of coproduction of IMP-4 and NDM-1 carbapenemases in hypermucoviscous *K. variicola*. The multiple resistance hybrid plasmid pNDM-IMP-1 possibly originates from *K. pneumoniae*, and ISCR1 element might contribute to the converge of $bla_{NDM-1}$ and $bla_{IMP-4}$ in this plasmid in *K. variicola* strain. Therefore, we must be vigilant regarding the outbreak and transmission of carbapenem-resistant *K. variicola* in children, especially for clinical isolates coproducing multiple carbapenemases. Long-time surveillance of these strains should be conducted as a priority.

Finally, we provided the experimental details as follows.

(i) Antimicrobial susceptibility testing and phenotypic analysis. The MICs of 18 antimicrobial agents, including cefotaxime, ceftazidime, cefepime, cefotetan, ertapenem, imipenem, meropenem, ciprofloxacin, cefoperazone-sulbactam (fixed concentration of sulbactam at 4 $\mu$g/mL), piperacillin–tazobactam (fixed

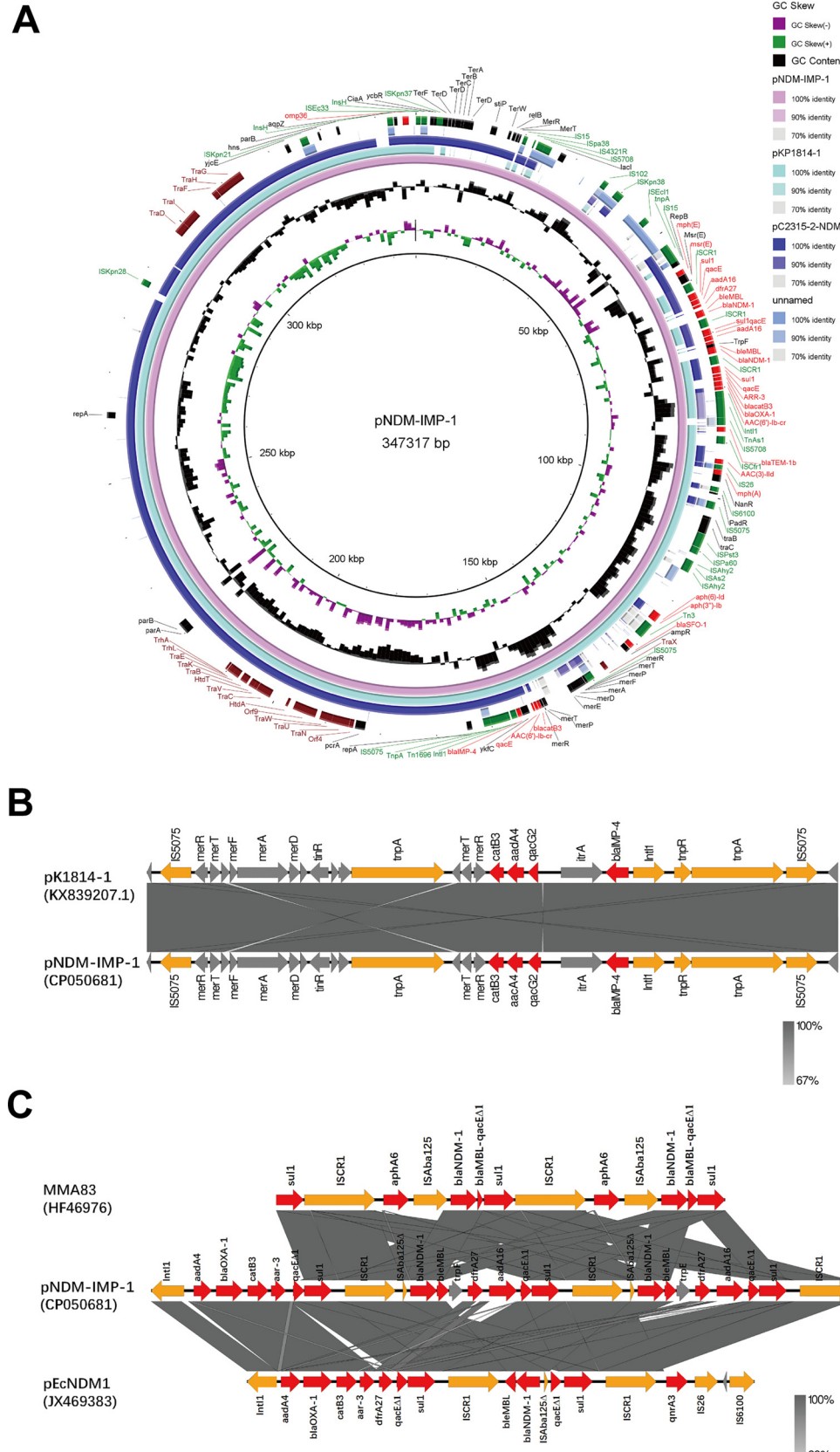

**FIG 2** Comparative genomics analysis of plasmid and genetic structure. (A) The circle genome alignment map of pNDM-IMP-1, pK1814-1 (KX839207.1), p2315-2-NDM (CP039829.1), and BHW35 unnamed (CP020508.1). The circle map

concentration of tazobactam at 4 $\mu$g/mL), amikacin, gentamicin, trimethoprim-sulfamethoxazole (1:19), aztreonam, ceftazidime-avibactam (fixed concentration of avibactam at 4 $\mu$g/mL), fosfomycin, colistin, and tigecycline, were detected using the broth microdilution method. The results were interpreted according to the Clinical and Laboratory Standards Institute (CLSI) guidelines (17), except for colistin and tigecycline, which were interpreted on the breakpoints of the European Committee on Antimicrobial Susceptibility Testing (EUCAST) (18) and the Food and Drug Administration (FDA) (19), respectively. *Escherichia coli* ATCC 25922 and *Pseudomonas aeruginosa* ATCC 27853 were used for quality control. NG-Test Carba 5 (NG Biotech, Guipry, France) was used for the identification of carbapenemase types. NG-Test CARBA 5 is a rapid diagnostic test based on the immunochromatographic detection of the five most common carbapenemase families (KPC, OXA-48-like, VIM, IMP, and NDM).

(ii) String test. String test was used to identify the HMV phenotype with a viscous string of >5 mm in length using a bacteriological loop to stretch the colony on blood agar plate at 37°C overnight.

(iii) Serum killing assay. Susceptibility to normal human serum was determined as described previously (20). Briefly, serum was obtained from healthy individuals and stored at −80°C. Bacterial cultures were grown to mid-log phase, and then 1 mL of culture was washed and resuspended in sterile normal saline (NS) to 0.5 McFarland turbidity standard suspension (mcf) and diluted 1:100. Then, 25 $\mu$L of the bacterial suspension was mixed with 75 $\mu$L of normal human serum. The mixtures were incubated at 37°C with shaking, and viable counts were obtained at 0, 1, 2, and 3 h by plating serial dilutions on Luria-Bertani (LB) agar plates after incubation for 18 to 24 h at 37°C. The response to serum killing in terms of viable counts was scored using six grades. A strain was classified as serum sensitive (grade 1 or 2), intermediately sensitive (grade 3 or 4), or serum resistant (grade 5 or 6). Each experiment was performed in triplicate.

(iv) Biofilm assay. The ability of *K. variicola* to form biofilms was tested as described previously (21). Overnight bacterial culture was diluted 1:100 in 5 mL of fresh LB medium. Then, 200 $\mu$L of the adjusted bacterial cultures was transferred to 96-well polystyrene microtiter plates, and an equivalent volume of LB was also tested. After incubation for 24 h at 37°C, the supernatant was discarded and the wells were washed three times with phosphate-buffered saline (PBS) and stained with 1% crystal violet for 15 to 30 min at room temperature. The bound dye was solubilized with 200 $\mu$L of absolute alcohol and quantified by measuring the absorbance at 590 nm. Three wells containing sterile LB without bacteria served as the negative control (value as Ac). Biofilm formation was scored relative to the absorbance value ($+++$, $\geq$4Ac; $++$, 2Ac to 4Ac; $+$, 1Ac to 2Ac, and $-$, $\leq$Ac). Each experiment was performed in triplicate.

(v) *Galleria mellonella* infection model. The virulence of *K. variicola* strain SHET-01 *in vivo* was tested in *G. mellonella* larvae as described previously (22). Bacterial cells were collected from overnight cultures and further adjusted with sterile normal saline to 1 × 10$^8$ CFU/mL. Ten larvae per group were injected at 10 $\mu$L per dose and then incubated at 37°C. One group injected with 10 $\mu$L of sterile normal saline was prepared as a negative control. The survival rates of *G. mellonella* were recorded every 12 h for 3 days. The hypervirulent *K. pneumoniae* strain NTUH-K2044 and the classic carbapenem-resistant *K. pneumoniae* strain KP1 from previous studies (randomly selected one strain from ST11 carbapenem-

**FIG 2** Legend (Continued)
was generated by BRIG tools. The locations of the genes on the plasmid circle map are indicated. Red represents the drug resistance gene, green represents the moving element, brown represents the plasmid conjugation related gene, and black represents other functional genes. (B and C) Comparative analysis of the genetic environment of *bla*$_{IMP-4}$ (B) and *bla*$_{NDM-1}$ (C). The intermediate gray area represents blast homology, and the genes in red, orange, and gray represent the drug-resistant genes, transfer elements and other genes, respectively.

resistant *K. pneumoniae* subset) (23) were used as hypervirulence controls and low virulence control, respectively. All experiments were carried out in duplicate or triplicate. Statistical analysis was performed and visualized using GraphPad Prism software.

(vi) **Genome sequencing and analysis.** The genomic DNA of strain SHET-01 was extracted and subjected to whole-genome sequencing using the PacBio Sequel single-molecule real-time (SMRT) sequencing platform and the Illumina NovaSeq sequencing platform (paired-end, 2 by 150 bp). Raw reads were *de novo* assembled using HGAP and CANU and were then modified using Pilon software. Annotation was performed using the RAST tool (https://rast.nmpdr .org/) and the NCBI Prokaryotic Genome Annotation Pipeline (PGAP) (http:// www.ncbi.nlm.nih.gov/genome/annotation_prok). The sequence type (ST) was determined using the database available at http://bigsdb.pasteur.fr/klebsiella/ klebsiella.html. Capsular typing was performed using Kaptive (https://github .com/katholt/Kleborate/). Virulence genes were identified using the virulence factor database (VFDB), and resistance genes were identified using the comprehensive antibiotic resistance database (CARD) (https://card.mcmaster.ca/) and the Resfinder database (https://cge.cbs.dtu.dk/services/). Plasmid replicons were determined using the PlasmidFinder tool (https://cge.cbs.dtu.dk/services/) and the pMLST tool (https://pubmlst.org/organisms/plasmid-mlst/). Insertion elements (IS) were identified using IS Finder (http://integrall.bio.ua.pt/).

The study protocol was approved by the Shanghai Children's Hospital Ethics Committee (Shanghai Jiao Tong University School of Medicine), and informed consent was waived.

**Data availability.** The complete sequence of SHET-01, including a chromosome and two plasmids, pNDM-IMP-1 and pNDM-IMP-2, has been deposited in the NCBI GenBank database with the accession numbers CP050680, CP050681, and CP050682, respectively.

## SUPPLEMENTAL MATERIAL

Supplemental material is available online only.
**SUPPLEMENTAL FILE 1**, PDF file, 0.6 MB.

## ACKNOWLEDGMENTS

We thank all members of the Clinical Laboratory of Shanghai Children's Hospital for their cooperation and technical help. This work was supported by Shanghai Municipal Key Clinical Specialty (grant number sshslczdzk06902).

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
