## [Reviewer comments · Microbiology Spectrum]

Microbiology Spectrum

Genetic characteristics and microbiological profile of hypermucoviscous multidrug-resistant *Klebsiella variicola* co-producing IMP-4 and NDM-1 carbapenemases

Bingjie Wang, Fen Pan, Dingding Han, Wantong Zhao, Yingying Shi, Yan Sun, Chun Wang, Tiandong Zhang, and Hong Zhang

Corresponding Author(s): Hong Zhang, Department of clinical laboratory, Shanghai Children's Hospital, Shanghai Jiaotong University, Shanghai, China

Review Timeline:

Submission Date:	September 16, 2021
Editorial Decision:	October 20, 2021
Revision Received:	December 10, 2021
Editorial Decision:	December 12, 2021
Revision Received:	December 13, 2021
Accepted:	December 15, 2021

Editor: Jeanette TEO

Reviewer(s): Disclosure of reviewer identity is with reference to reviewer comments included in decision letter(s). The following individuals involved in review of your submission have agreed to reveal their identity: Le Van Chuong (Reviewer #3)

Transaction Report:

DOI: <https://doi.org/10.1128/Spectrum.01581-21>

October 20, 2021

Prof. Hong Zhang

Department of clinical laboratory, Shanghai Children's Hospital, Shanghai Jiaotong University, Shanghai, China
No. 24, Lane 1400, West Beijing Road, Shanghai 200040, PR China
Shanghai
China

Re: Spectrum01581-21 (Genetic characteristics and microbiological portrait of hypermucoviscous extensively drug-resistant *Klebsiella variicola* co-producing IMP-4 and NDM-1 carbapenemase)

Dear Prof. Hong Zhang:

Dear Authors

The manuscript would be considered acceptable after conversion to shorter more concise format i.e. an Observations paper. Please refer to <https://journals.asm.org/journal/spectrum/article-types> for the article style and formatting. Briefly, the Observations format is limited to 1,200 words with a maximum of 2 figures and 25 references

Both reviewers have brought up two areas of concern, namely, the need to demonstrate the expression of dual carbapenemases, and the transferability of carbapenemase-bearing plasmid. I would like the authors to address the comments on plasmid expression and beta-lactamase production as that is the major premise of the title. Whereas I can overlook the requirements on the conjugation work and HMV associated features.

The English in the present form is not of publication quality and requires major improvement. Please carefully proof-read and spell check to eliminate grammatical errors

Other editorial comments:

line 86: SHER?

Line 89: confirmed not conformed

Line 88: Bruker Daltonics, not diatonic

Line 103: authors did manual microbroth dilution ? cefoperazone-sulbactam, piperacillin-tazobactam, amikacin, gentamicin, trimethoprim-sulfamethoxazole ratios are not the easiest to setup. Please clarify.

Line 119. What is mcf

Line 147. *K. pneumoniae* strain KP1 is low virulence strain control? Please provide a citation.

Fig.1 Can be removed.

Thank you for submitting your manuscript to Microbiology Spectrum. When submitting the revised version of your paper, please provide (1) point-by-point responses to the issues raised by the reviewers as file type "Response to Reviewers," not in your cover letter, and (2) a PDF file that indicates the changes from the original submission (by highlighting or underlining the changes) as file type "Marked Up Manuscript - For Review Only". Please use this link to submit your revised manuscript - we strongly recommend that you submit your paper within the next 60 days or reach out to me. Detailed information on submitting your revised paper are below.

Link Not Available

Sincerely,

Jeanette TEO

Journals Department
American Society for Microbiology

Reviewer comments:

Reviewer #1 (Comments for the Author):

Wang et al provide the characterization of an interesting *Klebsiella variicola* strain SHET-01 with hypermucoviscosity (HMV) phenotype and carrying two types of carbapenem resistance genes: blaIMP-4 and blaNDM-1. The authors report the results of antimicrobial susceptibility testing, biofilm, serum killing and *Galleria mellonella* infection, as well as WGS analysis of the strain. Although the strain carries many classes of antibiotic resistance genes, it remains susceptible to most tested antibiotics except beta-lactams. Furthermore, the authors show that the strain is avirulent to *G. mellonella*, even though the strain shows HMV phenotype and has a strong capacity to form biofilm. Finally, the WGS shows that the strain harbours two large plasmids, one of them (pNDM-IMP-1) carrying all plasmid-borne antibiotic genes.

The experimental design was well thought out and the data reported are of interest to the field. However it is not clear which carbapenemase gene or both genes contribute to the carbapenem resistance. The authors do not have the data of the expression of the two carbapenemase genes or the production of the enzymes. Given that the authors claim SHET-01 co-producing both IMP-4 and NDM-1 carbapenemases, the data of their expression or enzymatic activity should be demonstrated. Hypermucociscous *K. variicola* have been reported previously. In one such strain, the HMV was linked to a large plasmid (pKV8917). What is novel in this study is that the SHET-01 carries two large plasmids and one of them contains multiple carbapenem resistance genes. It would be interesting to see if the HMV phenotype of SHET-01 attributes to the MDR plasmid. Attempts should be made to test if pNDM-IMP-1 is able to conjugate to *K. variicola* and *K. pneumoniae*, and if the plasmid links to the rmpA-independent HMV.

Minor comments:

Line 104-105: reference for CLSI standard (year) should be given.

Line 106-107: which version (year) of the EUCAST and FDA breakpoints.

Line 116: reference for serum killing assay.

Line 127: check the reference numbers.

Line 140: reference for *G. mellonella* infection.

Line 172: sequence data should be made public.

Line 177: carbapenems are beta-lactams.

Line 179: An interpretation (e.g. S, I or R) for each antibiotic according to the breakpoints should be given in Table 1.

Line 183: Should add a short statement about what we see in Figure 1 in the figure legend.

Line 185: In Figure 2A, the data are from how many repeats and experiments?

Line 188: In Figure 2B, the scores of biofilm formation should also be shown on the figure.

Line 189: There are discrepancies between the method and Figure 3. The bacterial concentrations are from 10^4 , 10^5 , 10^6 and 10^7 cfu/ml in the method, but they are 10^5 , 10^6 , 10^7 and 10^8 cfu/ml in the figure.

Line 194: Are the data in Figure 3D the same from 3A, 3B and 3C? If yes, there should not be a separate panel.

Line 203: In Figure 4, what program was used to generate the circle map?

Line 218: Are there any similar between pNDM-IMP-1 (or pNDM-IMP-2) and pKV8917 from *K. variicola* (Rodríguez-Medina et al. 2020)?

Line 223: provide accession number of pKP1814-1.

Line 230: provide accession number of p2315-2-NDM.

Line 236: should be pNDM-IMP-1, not pNDM-IMP-4.

Line 284: HMV capsules in hvKp is regulated by rmpA, not encoded by rmpA.

Reviewer #3 (Comments for the Author):

Major points

1. Related to the title, the authors used the term "extensively drug-resistant-XDR" however in the result this strain was reported susceptibility to many class of antimicrobial agents except beta-lactams and its derivatives my question is that this isolate is truly XDR or just multi-drug resistant (MDR)? In addition, the term "co-producing IMP-4 and NDM-1 carbapenemase" should be also reconsidered, I agree that this strain possesses IMP-4 and NDM-1 genes on plasmid and it has phenotypic resistant to carpapenems but how could the authors make sure that it produces both carbapenemases?

2. to Antimicrobial susceptibility testing (line 98), Which version of CLSI did you use? As I understand, from CLSI 2017 up toward fosfomycin is used for testing and reporting of *E. coli* and *E. faecalis* urinary tract isolates only. Moreover, cefoperazone and cefoperazone-sulbactam are not the antimicrobial agents should be considered for routine testing and reporting. Could you

explain the reason why you did not test the cefoxitin? For quality control the utilizing of only E. coli ATCC 25922 is not enough, you should add P. aeruginosa ATCC 27853 for quality control of carbapenems. In 2017, a CLSI and EUCAST joint working group recommended broth microdilution (BMD), without surfactant, as the reference method for testing colistin (rBMD), I just wonder the broth microdilution method that you used is the same or not?

3. Line 153, could you explain the reason why you must be applied both sequencer platform for WGS?

Minor points

1. Should be added the citation: line 116; line 140; line 339
2. The format of citation should be edited: line 127; line 304; line 306
3. The first time mention should not in abbreviation form: VFDB line 161
4. Words should be corrected: SHER01 line 86; K. variicala line 260

Staff Comments:

Preparing Revision Guidelines

Please return the manuscript within 60 days; if you cannot complete the modification within this time period, please contact me. If you do not wish to modify the manuscript and prefer to submit it to another journal, please notify me of your decision immediately so that the manuscript may be formally withdrawn from consideration by Microbiology Spectrum.

To Editor:

1. The manuscript would be considered acceptable after conversion to shorter more concise format i.e. an Observations paper. Please refer to <https://journals.asm.org/journal/spectrum/article-types> for the article style and formatting. Briefly, the Observations format is limited to 1,200 words with a maximum of 2 figures and 25 references

Response: we have rewritten our manuscript in the format of observation paper.

2. Both reviewers have brought up two areas of concern, namely, the need to demonstrate the expression of dual carbapenemases, and the transferability of carbapenemase-bearing plasmid. I would like the authors to address the comments on plasmid expression and beta-lactamase production as that is the major premise of the title. Whereas I can overlook the requirements on the conjugation work and HMV associated features.

Response: thank you for your valuable comments, we have conducted experiments to give evidence to the expression of dual carbapenemases in Figure 1A in the revised version.

3. The English in the present form is not of publication quality and requires major improvement. Please carefully proof-read and spell check to eliminate grammatical errors

Response: thank you for your suggestion, a native English speaker has carefully checked and polished our manuscript.

4. Other editorial comments:

- A. line 86: SHER?

Response: we have deleted this mistake in the revised version.

- B. Line 89: confirmed not conformed

Response: we have deleted this mistake in the revised version.

- C. Line 88: Bruker Daltonics, not diatonic

Response: we have deleted this mistake in the revised version.

- D. Line 103: authors did manual microbroth dilution ? cefoperazone-sulbactam, piperacillin-tazobactam, amikacin, gentamicin, trimethoprim-sulfamethoxazole ratios are not the easiest to setup. Please clarify.

Response: yes, we had conducted antimicrobial susceptibility testing using the manual microbroth dilution, and we have added a description of the ratio of these antimicrobials in line 141-144 in the revise manuscript.

- E. Line 119. What is mcf

Response: mcf was the abbreviation of McFarland turbidity standard suspension, and we have added this description in line 158 in the revised version.

- F. Line 147. K. pneumoniae strain KP1 is low virulence strain control? Please provide a citation.

Response: this stain KP1 was randomly selected from our previous research. We have added this reference in line 185 in the revised version.

- G. Fig.1 Can be removed.

Response: we have deleted this figure in our revised version.

Review #1

1. Wang et al provide the characterization of an interesting *Klebsiella variicola* strain SHET-01 with hypermucoviscosity (HMV) phenotype and carrying two types of carbapenem resistance genes: blaIMP-4 and blaNDM-1. The authors report the results of antimicrobial susceptibility testing, biofilm, serum killing and *Galleria mellonella* infection, as well as WGS analysis of the strain. Although the strain carries many classes of antibiotic resistance genes, it remains susceptible to most tested antibiotics except beta-lactams. Furthermore, the authors show that the strain is avirulent to *G. mellonella*, even though the strain shows HMV phenotype and has a strong capacity to form biofilm. Finally, the WGS shows that the strain harbours two large plasmids, one of them (pNDM-IMP-1) carrying all plasmid-borne antibiotic genes.

The experimental design was well thought out and the data reported are of interest to the field. However it is not clear which carbapenemase gene or both genes contribute to the carbapenem resistance. The authors do not have the data of the expression of the two carbapenemase genes or the production of the enzymes. Given that the authors claim SHET-01 co-producing both IMP-4 and NDM-1 carbapenemases, the data of their expression or enzymatic activity should be demonstrated. Hypermucoviscous *K. variicola* have been reported previously. In one such strain, the HMV was linked to a large plasmid (pKV8917). What is novel in this study is that the SHET-01 carries two large plasmids and one of them contains multiple carbapenem resistance genes. It would be interesting to see if the HMV phenotype of SHET-01 attributes to the MDR plasmid. Attempts should be made to test if pNDM-IMP-1 is able to conjugate to *K. variicola* and *K. pneumoniae*, and if the plasmid links to the rmpA-independent HMV.

Response: Thank you for your valuable comments. We have added experiments to the revised manuscript to confirm the production of dual carbapenemases, as shown in Figure 1A.

About the idea that there might be some potential links between HMV phenotype and the MDR plasmid, this is a very interesting topic. We have compared this plasmid (pKV8917) with our plasmid, and found that their homology is very low. And we will sort this possibility out completely in the next research and conduct a more in-depth exploration of its HMV mechanism.

2. Minor comments:

- A. Line 104-105: reference for CLSI standard (year) should be given.

Response: we have added this reference in line 147 in the revised manuscript.

- B. Line 106-107: which version (year) of the EUCAST and FDA breakpoints.

Response: thank you for your advice, we have added these references in line 149-150 in the revised version.

- C. Line 116: reference for serum killing assay.

Response: we have added this reference in line 156 in the revised version.

- D. Line 127: check the reference numbers.

Response: we have corrected this reference in line 167 in the revised version.

- E. Line 140: reference for *G. mellonella* infection.

Response: we have added this reference in line 178 in the revised version.

- F. Line 172: sequence data should be made public.

Response: we have submitted our sequence data to the public database of NCBI GenBank and listed the assigned accession numbers in line 208-211 in the revised

manuscript.

- G. Line 177: carbapenems are beta-lactams.
Response: thank you for your reminder, we have corrected this description in line 86.
- H. Line 179: An interpretation (e.g. S, I or R) for each antibiotic according to the breakpoints should be given in Table 1.
Response: we have added an interpretation of testing antimicrobials in the Table S1 in the supplementary file.
- I. Line 183: Should add a short statement about what we see in Figure 1 in the figure legend.
Response: in the revised version manuscript, we have deleted this figure.
- J. Line 185: In Figure 2A, the data are from how many repeats and experiments?
Response: for all the phenotyping experiments, we conducted repeatedly three times.
- K. Line 188: In Figure 2B, the scores of biofilm formation should also be shown on the figure.
Response: thank you for your suggestion, we have added the biofilm formation scores of SHET-01 and NTUH-K2044 in the figure legend.
- L. Line 189: There are discrepancies between the method and Figure 3. The bacterial concentrations are from 10^4 , 10^5 , 10^6 and 10^7 cfu/ml in the method, but they are 10^5 , 10^6 , 10^7 and 10^8 cfu/ml in the figure.
Response: thank you for your reminder, we previously make a mistake in method, it actually is 10^5 , 10^6 , 10^7 and 10^8 cfu/ml. In the revised version, we have deleted the results at the concentrations of 10^5 , 10^6 , 10^7 cfu/ml in Figure 1 for avoidance of duplication.
- M. Line 194: Are the data in Figure 3D the same from 3A, 3B and 3C? If yes, there should not be a separate panel.
Response: we have improved this figure in Figure 1 in the revised version.
- N. Line 203: In Figure 4, what program was used to generate the circle map?
Response: we generated this map using the BRIG tools, and added this description in the figure legend of Figure S1 in supplementary file.
- O. Line 218: Are there any similar between pNDM-IMP-1 (or pNDM-IMP-2) and pKV8917 from *K. variceal* (Rodríguez-Medina et al. 2020)?
Response: as we mentioned above, the plasmid (pKV8917) shares very low homology with our plasmid of pNDM-IMP-1 and pNDM-IMP-2. Additionally, we found that pNDM-IMP-1 and pNDM-IMP-2 in our study were also not carried any genes which were hypothetically associated with the HMV of pKV8917 in their study. Therefore, we speculated that there might be some other unique mechanisms in our strain.
- P. Line 223: provide accession number of pKP1814-1.
Response: we have provided its accession number in line 91 in the revised manuscript.
- Q. Line 230: provide accession number of p2315-2-NDM.
Response: we have provided its accession number in line 99 in the revised manuscript.
- R. Line 236: should be pNDM-IMP-1, not pNDM-IMP-4.
Response: thank you for your reminder, we have corrected this description.
- S. Line 284: HMV capsules in hvKp is regulated by rmpA, not encoded by rmpA.
Response: thank you for your reminder, we have deleted this mistake.

Reviewer #3

Major points

1. Related to the title, the authors used the term "extensively drug-resistant-XDR" however in the result this strain was reported susceptibility to many class of antimicrobial agents except beta-lactams and its derivatives my question is that this isolate is truly XDR or just multi-drug resistant (MDR)? In addition, the term "co-producing IMP-4 and NDM-1 carbapenemase" should be also reconsidered, I agree that this strain possesses IMP-4 and NDM-1 genes on plasmid and it has phenotypic resistant to carbapenems but how could the authors make sure that it produces both carbapenemases?

Response: thank you for your valuable comments, we have reviewed our results of the antimicrobial susceptibility test, and this strain SHET-01 actually was multi-drug resistant. We have corrected this improper description. Additionally, about the production of dual carbapenemases, we have conducted experiment to confirm this using colloidal gold method (Figure 1A) in the revised version.

2. to Antimicrobial susceptibility testing (line 98), Which version of CLSI did you use? As I understand, from CLSI 2017 up toward fosfomycin is used for testing and reporting of *E. coli* and *E. faecalis* urinary tract isolates only. Moreover, cefoperazone and cefoperazone-sulbactam are not the antimicrobial agents should be considered for routine testing and reporting. Could you explain the reason why you did not test the ceftazidime? For quality control the utilizing of only *E. coli* ATCC 25922 is not enough, you should add *P. aeruginosa* ATCC 27853 for quality control of carbapenems. In 2017, a CLSI and EUCAST joint working group recommended broth microdilution (BMD), without surfactant, as the reference method for testing colistin (rBMD), I just wonder the broth microdilution method that you used is the same or not?

Response: Thank you for your valuable comments on the antimicrobial susceptibility testing. 1) We have used the CLSI M100 (2020) in our study and have added this reference in the revised manuscript. 2) In addition to urinary tract infections, fosfomycin has been increasingly used in the anti-infection of clinical multi-drug resistant Gram-negative bacterial in recent years, which *in vitro* susceptibility usually refer to the interpretation criteria of urinary tract *Escherichia coli* and *Enterococcus faecalis*. However, in our study, considering the strain SHET-01 was isolated from a pediatric patient, and fosfomycin was not recommended used in children in China, we have deleted the data of this antimicrobial sensitivity. 3) In China, cefoperazone and cefoperazone-sulbactam were widely used in the clinical anti-infections, need provide their data to give a basis for clinical treatment. About ceftazidime, we have not included this antimicrobial in our antimicrobial susceptibility testing, because we had tested another cephamycin antibiotic, ceftazidime. 4) Actually, we also had used *P. aeruginosa* ATCC 27853 as quality control, and have added this description in line 150-151 in the revised version. 5) Colistin susceptibility test was performed using polystyrene plastic plate without tween-80 in our study.

3. Line 153, could you explain the reason why you must be applied both sequencer platform for WGS?

Response: PacBio Sequel single-molecule real-time (SMRT) sequencing platform was a third-generation sequencing platform, and the Illumina NovaSeq sequencing platform was based on second-generation sequencing technologies. The second-generation sequencing technologies have offered vast improvements over Sanger sequencing, their limitations, especially their short-read lengths (50-500bp), make them poorly suited for some particular biological problems, including assembly and determination of complex genomic regions, gene isoform detection, and methylation detection. The third-generation PacBio sequencing is a method for real-time sequencing, and it offers much longer read lengths (> 10kb) and faster runs than second-generation sequencing methods but is hindered by a lower throughput and higher error rate (~15%)[1]. Since the advantages of third-generation sequencing and second-generation sequencing are complementary, we therefore used hybrid-sequencing strategies that make use of both technologies to overcome the disadvantages of each alone.

Reference

1. Rhoads A, Au KF. PacBio Sequencing and Its Applications. *Genomics Proteomics Bioinformatics*. 2015 Oct;13(5):278-89.

4. Minor points

A. Should be added the citation: line 116; line 140; line 339

Response: we have added references in corresponding lines.

B. The format of citation should be edited: line 127; line 304; line 306

Response: we have corrected these references.

C. The first time mention should not in abbreviation form: VFDB line 161

Response: we have added its full name in line 199 in the revised version.

D. Words should be corrected: SHER01 line 86; K. variicala line 260

Response: thank you for your careful check, we have corrected these mistakes.

December 12, 2021

Prof. Hong Zhang

Department of clinical laboratory, Shanghai Children's Hospital, Shanghai Jiaotong University, Shanghai, China
No. 24, Lane 1400, West Beijing Road, Shanghai 200040, PR China
Shanghai
China

Re: Spectrum01581-21R1 (Genetic characteristics and microbiological profile of hypermucoviscous multidrug-resistant *Klebsiella variicola* co-producing IMP-4 and NDM-1 carbapenemases)

Dear Prof. Hong Zhang:

Thank you for submitting your manuscript to Microbiology Spectrum. As you will see your paper is very close to acceptance. Please modify the manuscript along the lines I have recommended. As these revisions are quite minor, I expect that you should be able to turn in the revised paper in less than 30 days, if not sooner. If your manuscript was reviewed, you will find the reviewers' comments below.

Required amendments:

-Line 56. Is this carb 5 lateral flow assay ? if so, I would like you to replace the sentence with " Consistent with this, the coproduction of NDM-1 and IMP-4 carbapenemases was confirmed by NG-Test® CARBA 5 lateral flow assay (Figure 1A)"

-Under experimental methods, after line 138. Can you add the description of how carb5 works "NG-Test® CARBA 5 is a lateral flow assay NG-Test Carba 5 (NG Biotech, Guipry, France). It is a rapid diagnostic test (less than or equal to)15 min) based on the immunochromatographic detection of the five most common carbapenemase families (KPC, OXA-48-like, VIM, IMP, and NDM) directly from bacterial colonies."

-Line 297. Edit sentence to "The production of NDM and IMP carbapenemases in SHET-01 detected on a NG-Test® CARBA 5."

When submitting the revised version of your paper, please provide (1) point-by-point responses to the issues I raised in your cover letter, and (2) a PDF file that indicates the changes from the original submission (by highlighting or underlining the changes) as file type "Marked Up Manuscript - For Review Only". Please use this link to submit your revised manuscript. Detailed instructions on submitting your revised paper are below.

Link Not Available

Sincerely,

Jeanette TEO

Reviewer comments:

Preparing Revision Guidelines

To submit your modified manuscript, log onto the eJP submission site at <https://spectrum.msubmit.net/cgi-bin/main.plex>. Go to Author Tasks and click the appropriate manuscript title to begin the revision process. The information that you entered when you

first submitted the paper will be displayed. Please update the information as necessary. Here are a few examples of required updates that authors must address:

- point-by-point responses to the issues I raised in your cover letter
- Upload a compare copy of the manuscript (without figures) as a "Marked-Up Manuscript" file.
- Each figure must be uploaded as a separate file, and any multipanel figures must be assembled into one file.
- Manuscript: A .DOC version of the revised manuscript
- Figures: Editable, high-resolution, individual figure files are required at revision, TIFF or EPS files are preferred

Please return the manuscript within 60 days; if you cannot complete the modification within this time period, please contact me. If you do not wish to modify the manuscript and prefer to submit it to another journal, please notify me of your decision immediately so that the manuscript may be formally withdrawn from consideration by Microbiology Spectrum.

Required amendments:

-Line 56. Is this carb 5 lateral flow assay ? if so, I would like you to replace the sentence with

" Consistent with this, the coproduction of NDM-1 and IMP-4 carbapenemases was confirmed by NG-Test® CARBA 5 lateral flow assay (Figure 1A)"

Response: yes, it is carb 5 lateral flow assay, and we have modified this sentence as you suggested in line 56 in the revised manuscript.

-Under experimental methods, after line 138. Can you add the description of how carb5 works

"NG-Test® CARBA 5 is a lateral flow assay NG-Test Carba 5 (NG Biotech, Guipry, France). It is a rapid diagnostic test (less than or equal to 15 min) based on the immunochromatographic detection of the five most common carbapenemase families (KPC, OXA-48-like, VIM, IMP, and NDM) directly from bacterial colonies."

Response: we have added these descriptions in line 151-155 in the revised manuscript.

-Line 297. Edit sentence to "The production of NDM and IMP carbapenemases in SHET-01 detected on a NG-Test® CARBA 5."

Response: we have modified this sentence as you required in line 297-298 in the revised version.

December 15, 2021

Prof. Hong Zhang

Department of clinical laboratory, Shanghai Children's Hospital, Shanghai Jiaotong University, Shanghai, China
No. 24, Lane 1400, West Beijing Road, Shanghai 200040, PR China
Shanghai
China

Re: Spectrum01581-21R2 (Genetic characteristics and microbiological profile of hypermucoviscous multidrug-resistant *Klebsiella variicola* co-producing IMP-4 and NDM-1 carbapenemases)

Dear Prof. Hong Zhang:

Your manuscript has been accepted, and I am forwarding it to the ASM Journals Department for publication. You will be notified when your proofs are ready to be viewed.

Sincerely,

Jeanette TEO
Editor, Microbiology Spectrum
